# Policy-Conditioned Uncertainty Sets for Robust Markov Decision Processes

**Andrea Tirinzoni**
Politecnico di Milano
andrea.tirinzoni@polimi.it

**Xiangli Chen**
Amazon Robotics
cxiangli@amazon.com

**Marek Petrik**
University of New Hampshire
mpetrik@cs.unh.edu

**Brian D. Ziebart**
University of Illinois at Chicago
bziebart@uic.edu

## Abstract

What policy should be employed in a Markov decision process with uncertain parameters? Robust optimization's answer to this question is to use rectangular uncertainty sets, which independently reflect available knowledge about each state, and then to obtain a decision policy that maximizes the expected reward for the worst-case decision process parameters from these uncertainty sets. While this rectangularity is convenient computationally and leads to tractable solutions, it often produces policies that are too conservative in practice, and does not facilitate knowledge transfer between portions of the state space or across related decision processes. In this work, we propose non-rectangular uncertainty sets that bound marginal moments of state-action features defined over entire trajectories through a decision process. This enables generalization to different portions of the state space while retaining appropriate uncertainty of the decision process. We develop algorithms for solving the resulting robust decision problems, which reduce to finding an optimal policy for a mixture of decision processes, and demonstrate the benefits of our approach experimentally.

## 1 Introduction

Policies with high expected reward are often desired for uncertain decision processes with which little experience exists. Specifically, we consider the setting in which only a limited number of trajectories from a sub-optimal control policy through a decision process are available. Robust control approaches for this task [1, 2, 3, 4] define uncertainty sets for the decision process based on the limited outcome samples and seek the policy that maximizes this expected reward for the worst possible choice of decision process parameters in these sets.

When the uncertainty sets relating to different decision process states are jointly constrained in seemingly natural ways, the robust control problem becomes NP-hard (e.g., [5, 6]). To avoid these computationally intractable robust control problems, uncertainty sets have often been independently constructed for parameters associated with a particular state-action pair or particular state—$s, a$-rectangularity or $s$-rectangularity [7, 8, 4], respectively. Unfortunately, independently assuming the worst-case in every encountered state is often too conservative in practice to be useful [9].

Leveraging ideas from distributionally robust optimization [10, 11, 12], we construct *policy-conditioned marginal uncertainty sets* for robustly learning a decision policy that optimizes the reward given trajectory samples produced by a sub-optimal policy. State transition dynamics under our formulation are estimated based on two competing objectives. First, the estimated dynamics must

(approximately) match measured properties observed under the sub-optimal reference policy. Second, the estimated dynamics must be the worst case for the simultaneously-hypothesized optimal policy.

This formulation has three main benefits: (1) *Non-rectangularity:* Our uncertainty sets are defined by feature-based statistics of distributions over entire trajectories, enabling generalization across states; (2) *Off-policy robustness:* We define our performance objective using the desired control policy and the uncertainty set using the sub-optimal data generation policy; and (3) *Convex parameter optimization:* We avoid the nonconvex parameter optimization pitfalls of other nonrectangular formulations by shifting the main computational difficulties to parameterized prediction/control problems (which can be efficiently approximated). Together, these properties aid in addressing a number of existing concerns for robust control, including settings in which the state definition violates the Markov assumption [13] or the transition probabilities are derived from limited data sets [3, 9].

In the remainder of this paper, we review existing robust control methods and directed information theory concepts in Section 2. Using these concepts, we formulate the robust control task using feature-based marginal constraints in Section 3. We reformulate this problem and present algorithms for solving it using a combination of convex optimization and dynamic programming to optimize a non-Markovian mixed decision process optimal control problem that arises from the formulation. We evaluate our approach in Section 4 to demonstrate its comparative benefits over rectangular robust control methods. Lastly, we provide concluding thoughts and discuss future work in Section 5.

## 2 Background and Related Work

### 2.1 Robust control

The Markov Decision Process (MDP) with state set $\mathcal{S}$ and action set $\mathcal{A}$ povides a common formulation of discrete control problems. In the MDP, the transition probabilities are given by $\tau(s_{t+1} \mid s_t, a_t)$ and the reward is $R(s_t, a_t, s_{t+1})$. Though consideration is often restricted to deterministic Markovian policies, $\pi : \mathcal{S} \to \mathcal{A}$, the generalization to *randomized* Markovian policies $\Pi_M = \{\pi : \mathcal{S} \to \Delta_{\mathcal{A}}\}$ provides stochastic mappings from the current state to actions. Even more generally, we will consider non-Markovian, history-dependent, randomized policies $\Pi_H = \{\pi : \mathcal{S}^t \times \mathcal{A}^t \to \Delta_{\mathcal{A}}\}$ in this work.

The expected sum of rewards or return $\rho$ of a policy $\pi$ applied to an MDP with dynamics $\tau$ and reward function $R$ is: $\rho_R(\pi, \tau) = \mathbb{E}_{\tau,\pi}[\sum_{t=1}^{T-1} R(S_t, A_t, S_{t+1})]$. For decision problems, the standard objective is to choose a policy that maximizes the expected sum of rewards: $\max_\pi \rho_R(\pi, \tau)$. Since a Markovian and deterministic policy always exists that maximizes this quantity, one with those characteristics is typically sought when solving this optimization problem by many well-known algorithms, such as value iteration or policy iteration [14].

Unfortunately, in many settings the dynamics $\tau$ are not entirely known. Control policies are needed that can perform well despite this uncertainty about the decision process. One option is to formally define the uncertainty as a set of possibilities and assume the worst case (Definiton 1).

**Definition 1.** *The robust control problem is to find a control policy $\pi \in \Pi$ that performs best for the worst-case choice of state transition dynamics, $\tau \in \Xi$:*

$$\max_{\pi \in \Pi} \min_{\tau \in \Xi} \rho(\pi, \tau) = \max_{\pi \in \Pi} \min_{\tau \in \Xi} \mathbb{E}_{\tau,\pi} \left[ \sum_{t=1}^{T-1} R(S_t, A_t, S_{t+1}) \right]. \tag{1}$$

The specification of the uncertainty set(s), $\Xi$, has significant implications for the tractability of this problem. Robust MDPs [7] are typically used to represent uncertainty in transition probabilities and rewards in regular MDPs. When the state-transition probabilities for different states are jointly constrained in arbitrary ways, the robust control problem becomes NP-hard [5, 6]. Two common forms of constraints that enable efficient solutions are *s,a*-rectangular and *s*-rectangular [4] constraint sets. This form arises when transition probabilities are not known precisely, but are known to be bounded in terms of an $L_1$ norm. A corresponding robust MDP has uncertain transition probabilities:

$$\Xi = \{\tau \ : \ \forall s, a \in \mathcal{S} \times \mathcal{A}, \ \|\tau(\cdot|s,a) - p(\cdot \mid s,a)\|_1 \leq c\}.$$

This is an $s, a$-rectangular set. It employs independent constraints for each state-action pair or state ($s$-rectangular set). A convenient way to model a robust MDP is to introduce a set of outcomes $\mathcal{B}$ to represent the uncertainty in transitions and rewards. The transition probabilities are then defined

as $p(s_{t+1} \mid s_t, a_t, b_t)$ and rewards become $r(s_t, a_t, b_t, s_{t+1})$, while $\xi(b_t|s_t, a_t)$ denotes the nature's policy, i.e., a distribution over outcomes.

The optimal value function $v^\star$ in a robust MDP with $s$-rectangular and $s, a$-rectangular uncertainty sets (and discount factor $\gamma$) satisfies the Bellman optimality equation for each $s \in \mathcal{S}$ as follows:

$$v^\star(s) = \max_{\pi \in \Pi} \min_{\xi \in \Xi} \sum_{a \in \mathcal{A}} \sum_{b \in \mathcal{B}} \pi(a|s)\xi(b|s, a)\Big(r(s, a, b, s') + \gamma \sum_{s' \in \mathcal{S}} p(s' \mid s, a, b)v^\star(s')\Big). \quad (2)$$

In our formulation, we consider state-action feature-based constraints over the marginals of state-action sequences to define our uncertainty sets. When the sum of rewards and the constraints are defined in terms of different policies, this naturally induces a "belief state" that is similar to the augmenting set of outcomes $\mathcal{B}$ previously described. In our case, this augmenting information tracks the relative significance of the policies for providing robustness based on the sum of rewards versus matching feature-based measurements from training trajectories.

## 2.2 Directed information theory for processes

We make extensive use of ideas and notation from directed information theory [15, 16, 17, 18, 19]. Under this theory, processes—the products of $T$ conditional probabilities over a sequence of $T$ variables—are treated as first-order objects. The causally conditioned probability distribution [20], $p(\mathbf{y}_{1:T}||\mathbf{x}_{1:T}) \triangleq \prod_{t=1}^{T} p(y_t|\mathbf{y}_{1:t-1}, \mathbf{x}_{1:t})$, illustrates the notation for this process of generating the sequence of $\mathbf{y}_{1:T}$ variables given the sequence of $\mathbf{x}_{1:T}$ variables. It differs from the conditional probability distribution, $p(\mathbf{y}_{1:T}|\mathbf{x}_{1:T}) = \prod_{t=1}^{T} p(y_t|\mathbf{x}_{1:T}, \mathbf{y}_{1:t-1})$, in the limited history of $x$ variables each $y_t$ variable is conditioned upon.

Both (stochastic) control policies, $\pi(\mathbf{a}_{1:T}||\mathbf{s}_{1:T}) \triangleq \prod_{t=1}^{T} \pi(a_t|\mathbf{a}_{1:t-1}, \mathbf{s}_{1:t})$, and (stochastic) state transition dynamics, $\tau(\mathbf{s}_{1:T}||\mathbf{a}_{1:T-1}) \triangleq \prod_{t=1}^{T} \tau(s_t|\mathbf{s}_{1:t-1}, \mathbf{a}_{1:t-1})$, can be expressed using this notation. The joint probability distribution over states and actions is then $p(\mathbf{a}_{1:T}, \mathbf{s}_{1:T}) = \pi(\mathbf{a}_{1:T}||\mathbf{s}_{1:T})\tau(\mathbf{s}_{1:T}||\mathbf{a}_{1:T-1})$, and the expected reward can be expressed as an affine combination of bilinear functions of these processes:

$$\rho_R(\pi, \tau) = \sum_{\mathbf{a}_{1:T}} \sum_{\mathbf{s}_{1:T}} \pi(\mathbf{a}_{1:T}||\mathbf{s}_{1:T})\tau(\mathbf{s}_{1:T}||\mathbf{a}_{1:T-1}) \sum_{t=1}^{T-1} R(s_t, a_t, s_{t+1}). \quad (3)$$

Additionally, the uncertainty of state sequence outcomes can be quantified using the causally conditioned entropy:

$$H_{\tau, \pi}(S_{1:T}||A_{1:T-1}) = - \sum_{\mathbf{a}_{1:T}, \mathbf{s}_{1:T}} \pi(\mathbf{a}_{1:T}||\mathbf{s}_{1:T})\tau(\mathbf{s}_{1:T}||\mathbf{a}_{1:T-1}) \log \tau(\mathbf{s}_{1:T}||\mathbf{a}_{1:T-1}). \quad (4)$$

Of crucial importance for optimization purposes, the set of causally conditioned probability distributions is convex and the causal entropy is a convex function of those probabilities [21].

# 3 Marginally-Constrained Robust Control Processes

We define constraints on uncertainties about a decision process based on its interactions with a reference policy. In other words, state-action trajectories through the decision process are available that were produced from a policy that may be quite different from the optimal one. Similarly to previous works [5, 22], we propose practical algorithms for this problem by augmenting the state space.

## 3.1 Defining Uncertainty Sets with Marginal Features

We consider a feature function $\phi : \mathcal{S} \times \mathcal{A} \times \mathcal{S} \to \mathbb{R}^d$ characterizing the relationships between states and actions to restrict the set of possible realizations of uncertain MDP parameters. We denote the first moment of the occupancy frequencies with respect to $\phi$ (also known as feature expectations in the inverse reinforcement learning literature [23, 24]) as $\boldsymbol{\kappa}_\phi(\pi, \tau) := \mathbb{E}_{\tau, \pi}\left[\sum_{t=1}^{T-1} \phi(S_t, A_t, S_{t+1})\right]$, while we denote the empirical sample statistics, which are measured from $N$ sample trajectories,

as $\widehat{\boldsymbol{\kappa}} = \frac{1}{N} \sum_{i=1}^{N} \sum_{t=1}^{T-1} \phi(s_t^{(i)}, a_t^{(i)}, s_{t+1}^{(i)})$. Based on these quantities, we can now define the robust control problem with constraints using marginal statistics of the state-action sequence to define the uncertainty set $\Xi$.

**Definition 2.** *The marginally-constrained robust control problem given reference policy $\tilde{\pi}$ is:*

$$\max_{\pi \in \Pi} \min_{\tau \in \Xi} \rho(\pi, \tau) - \frac{1}{\lambda} H_{\tau, \bar{\pi}}(S_{1:T} || A_{1:T-1}), \qquad (5)$$

*where $\Xi$ is the set of all transition probabilities whose feature expectations match the empirical sample statistics, i.e., $\Xi = \{\tau \mid \boldsymbol{\kappa}_\phi(\tilde{\pi}, \tau) = \widehat{\boldsymbol{\kappa}}\}$. In general, and of practical significance, slack can also be added to the constraints, leading to a relaxed uncertainty set $\widetilde{\Xi} = \{\tau \mid ||\boldsymbol{\kappa}_\phi(\tilde{\pi}, \tau) - \widehat{\boldsymbol{\kappa}}|| \leq \beta\}$[1]. We include an optional causal entropy (Equation 4) regularization penalty term, $\frac{1}{\lambda} H_{\tau, \bar{\pi}}(S_{1:T} || A_{1:T-1})$, where $\lambda \in (0, \infty)$ is a provided parameter and $\bar{\pi}(\mathbf{a}_{1:T} || \mathbf{s}_{1:T})$ is an arbitrary distribution.*

Intuitively, our formulation allows constraints for whole trajectories rather single state-action pairs, as with rectangular constraints. Furthermore, features $\phi$ allow us to specify properties of the unknown transition dynamics that generalize globally across the state-action space, which is not possible using local constraints, such as rectangular ones. When limited data is available and generalization is therefore required to achieve good performance, this constitutes a significant advantage. Finally, our optional entropy regularization term leads to smoother solutions, where the smoothness is controlled by parameter $\lambda$. Many previous works have shown the benefits of having entropy-based smoothing [2, 25].

In practice, the design of the feature function $\phi$ is fundamental for properly constraining the estimated transition probabilities. Although a specific choice is highly application dependent, the features should in general encode known properties of the underlying MDP. Since our solution reduces to finding dynamics that induce a behavior on the reference policy, specified through $\boldsymbol{\kappa}_\phi$, that approximately matches the one observed from the given trajectories, many analogies exist with feature design in the IRL literature (see, e.g., chapter 6 of [26]). Common choices thus include indicator functions over important properties/events, such as reaching certain goal states, entering dangerous zones, taking very likely (or unlikely) transitions, and so on. The key consequence of adding these kinds of features is that the probability of these events occurring under the estimated dynamics will be (approximately) the same as the one observed in the given trajectories. Consider, for instance, an MDP where $s, a, s'$ triples with some known property $\mathcal{P}(s, a, s')$ have zero probability (e.g., in a gridworld or a chain-walk domain, a transition is impossible if $s$ and $s'$ are not adjacent). Then, using a feature $\phi(s, a, s') = \mathbb{1}[\mathcal{P}(s, a, s')]$, i.e., an indicator function over $\mathcal{P}$, will constrain the estimated transition probabilities to be zero for all triples where such property holds. In fact, $\kappa_\phi(\tau, \tilde{\pi}) = 0$ and $\hat{\kappa}_\phi = 0$ for any reference policy. More generally, most MDPs of practical interest have properties that couple the transition probabilities of several state-action pairs. Capturing these global properties using moment-based constraints is typically much better than focusing on single states or state-action pairs, which is more prone to overfitting the given trajectories. In the limiting case, one could consider a separate feature (e.g., an indicator) over each $s, a, s'$ triple. However, similarly to rectangular solutions, having separate constraints for different state-action pairs is likely to lead to very conservative solutions in the presence of limited data. Finally, notice that using an indicator function over each $s, a, s'$ triple is equivalent to matching the (empirical) joint distribution $p(S_t, A_t, S_{t+1})$ induced by the reference policy and the true dynamics. Thus, even when we consider a different constraint for each triple, our solution implicitly couples the transition probabilities of different state-action pairs and differs from a rectangular formulation which focuses on matching the conditional distribution $p(S_{t+1}|S_t, A_t)$.

A key characteristic of this formulation is the difference in control policies: the expected reward is defined in terms of $\pi$, while the constraints are defined in terms of $\tilde{\pi}$. Unfortunately, treating the marginally-constrained robust control problem (Definition 1) as an optimization problem over the individual state transition probabilities, $\tau(s_{t+1}|s_t, a_t)$, appears daunting. This is because the constraints in Equation (5) are not convex functions of those transition probabilities. We instead consider optimizing the control policy and state transition dynamics as causally conditioned probability distributions in the following section. Though the solution for this formulation does not naturally have a Markovian property, our process estimation leads to an augmented-Markovian representation in Section 3.3.

## 3.2 Reformulation as Process Estimation

We re-express the optimization problem of Definition 2 using processes—the causally conditioned probabilities of Section 2.2—for the control policy $\pi(a_{1:T-1}||s_{1:T-1})$ and state transition dynamics $\tau(s_{1:T}||a_{1:T-1})$, which conveniently combine the individual conditional probabilities over the state-action sequence. Notice that we consider stochastic processes ending with a state at time $T$ and an action at time $T-1$. Using this new notation, we now reformulate our main optimization problem in a more convenient manner.

**Theorem 1.** *The marginally-constrained robust control problem of Definition 2 can be solved by posing it as an unconstrained zero-sum game parameterized by a vector of Lagrange multipliers, $\boldsymbol{\omega}$:*

$$\max_{\boldsymbol{\omega} \in \mathbb{R}^d} \max_{\pi \in \Pi} \operatorname*{softmin}_{\tau \in \Xi} \left( \mathbb{E}_{\tau, \pi} \left[ \sum_{t=1}^{T-1} R(S_t, A_t, S_{t+1}) \right] + \mathbb{E}_{\tau, \widetilde{\pi}} \left[ \sum_{t=1}^{T-1} \boldsymbol{\omega} \cdot \boldsymbol{\phi}(S_t, A_t, S_{t+1}) \right] \right) - \boldsymbol{\omega} \cdot \widehat{\boldsymbol{\kappa}}, \quad (6)$$

*where* $\operatorname{softmin}_{x \in \mathcal{X}} f(x) = -\frac{1}{\lambda} \log \sum_{x \in \mathcal{X}} e^{-\lambda f(x)}$ *and* $\cdot$ *denotes the dot product.*

The proof is given in Appendix A. Notice that Theorem 1 holds for the slack-free uncertainty set $\Xi$ of Definition 2. Using the slack-based version leads to regularization of the dual parameters $\boldsymbol{\omega}$. As shown by [27], adding $l_1$ regularization $-\beta||\boldsymbol{\omega}||_1$ to the dual objective is equivalent to a constraint $||\boldsymbol{\kappa}_{\boldsymbol{\phi}}(\widetilde{\pi}, \tau) - \widehat{\boldsymbol{\kappa}}||_1 \leq \beta$ in the primal, while adding $l_2^2$ regularization $-\frac{\alpha}{2}||\boldsymbol{\omega}||_2^2$ is equivalent to an $l_2^2$ potential on the constraint values in the primal. In practice, it is important to add $l_1$ and/or $l_2^2$ regularization to ensure proper convergence of the algorithm. Both types of regularization enjoy similar theoretical guarantees [28].

We now address the inner minimax game for choosing $\tau$ and $\pi$ in Section 3.3 and the outer optimization of $\boldsymbol{\omega}$ from Equation (6) in Section 3.4.

## 3.3 Mixed Objective Minimax Optimal Control

Choosing state transition dynamics to optimize a mixture of expected returns under different control policies, $\pi$ and $\tilde{\pi}$ (Definition 3)[2] is an important subproblem arising from our formulation of robust control as a process estimation task with robustness properties and uncertainty sets defined by different control policies. To the best of our knowledge, this problem has not been previously investigated in the literature.

**Definition 3.** *Given two control policies $\pi$ and $\tilde{\pi}$, and two reward functions $R$ and $\tilde{R}$, the mixed objective optimization problem seeks state transition dynamics $\tau$ that minimizes a mixture of these weighted by $\theta \geq 0$: $\min_\tau \{\theta \rho_R(\pi, \tau) + (1-\theta)\rho_{\tilde{R}}(\tilde{\pi}, \tau)\}$.*

Notice that the inner minimization of Equation (6) is an entropy-regularized instance of this problem. In fact, we can set $\widetilde{R}(s_t, a_t, s_{t+1}) \leftarrow \boldsymbol{\omega} \cdot \boldsymbol{\phi}(s_t, a_t, s_{t+1})$ and $\theta = \frac{1}{2}$ (provided that rewards are properly rescaled). As we already know from Theorem 1, the entropy leads to a softmin solution and does not pose any additional complication in solving the optimization problem of Definition 3. Furthermore, in the inner zero-sum game of Equation (6), $\pi$ is chosen as the maximizer of $\rho(\pi, \tau)$. Thus, we can see Definition 3 as a special case where $\pi$ is fixed rather than chosen dynamically.

An important observation for this problem is that the optimal transition dynamics are not Markovian. Indeed, the influence of $\rho_R$ and $\rho_{\tilde{R}}$ on choosing the next-state distribution at some decision point depends on how probable it is for that decision point to be realized under $\pi$ and under $\tilde{\pi}$. This, in turn, depends on the entire history of states and actions leading to the current decision point. However, we establish that this non-Markovian problem can be Markovianized by augmenting the current state-action pair with a continuous "belief state" as follows:

$$b(\mathbf{a}_{1:t}||\mathbf{s}_{1:t}) \triangleq \frac{\prod_{i=1}^t \pi(a_i|\mathbf{a}_{1:i-1}, \mathbf{s}_{1:i})}{\prod_{i=1}^t \pi(a_i|\mathbf{a}_{1:i-1}, \mathbf{s}_{1:i}) + \prod_{i=1}^t \widetilde{\pi}(a_i|\mathbf{a}_{1:i-1}, \mathbf{s}_{1:i})} = \frac{\pi(\mathbf{a}_{1:t}||\mathbf{s}_{1:t})}{\pi(\mathbf{a}_{1:t}||\mathbf{s}_{1:t}) + \widetilde{\pi}(\mathbf{a}_{1:t}||\mathbf{s}_{1:t})}. \quad (7)$$

The belief state tracks the relative probability of the decision point under $\pi$ and $\tilde{\pi}$. Defining it in this manner is convenient because it limits the domain for $b$ to $[0, 1]$. It can also be updated to incorporate

a new action $a_{t+1}$ in state $s_{t+1}$ as:

$$b(\mathbf{a}_{1:t+1}||\mathbf{s}_{1:t+1}) = \frac{b(\mathbf{a}_{1:t}||\mathbf{s}_{1:t})\pi(a_{t+1}|\mathbf{a}_{1:t}, \mathbf{s}_{1:t+1})}{b(\mathbf{a}_{1:t}||\mathbf{s}_{1:t})\pi(a_{t+1}|\mathbf{a}_{1:t}, \mathbf{s}_{1:t+1}) + (1 - b(\mathbf{a}_{1:t}||\mathbf{s}_{1:t}))\widetilde{\pi}(a_{t+1}|\mathbf{a}_{1:t}, \mathbf{s}_{1:t+1})}. \quad (8)$$

Augmenting with the belief state of Equation (7), we prove that it is possible to compute a Markovian solution to the inner zero-sum game of Equation (6) and, thus, to the optimization problem of Definition 3.

**Theorem 2.** *Let $\tilde{\pi}$ be a given randomized Markovian policy and $Z(s_t, a_t, b_{t-1}) = b_{t-1} + (1 - b_{t-1})\tilde{\pi}(a_t|s_t)$, where $b_t$ is the belief state defined in Equation (7). Then, a solution $(\pi^*, \tau^*)$ to the inner zero-sum game of Equation (6) is:*

$$\tau^*(s_{t+1}|s_t, a_t, b_t) = \frac{e^{-\lambda Q(s_t, a_t, b_t, s_{t+1})}}{\sum_{s'} e^{-\lambda Q(s_t, a_t, b_t, s')}}; \; \pi^*(s_t, b_{t-1}) = \underset{a_t}{\operatorname{argmax}} \, Q_R\left(s_t, a_t, \frac{b_{t-1}}{Z(s_t, a_t, b_{t-1})}\right), \quad (9)$$

*with $Q$ as the value of a transition to state $s_{t+1}$, $V$ as the value of state $s_t$ and belief state $b_{t-1}$, and $Q_R$ as the expected return from $R$ obtained by taking action $a_t$ in state $s_t$ and belief state $b_t$:*

$$Q(s_t, a_t, b_t, s_{t+1}) = b_t R(s_t, a_t, s_{t+1}) + (1 - b_t)\widetilde{R}(s_t, a_t, s_{t+1}) + V(s_{t+1}, b_t), \quad (10)$$

$$V(s_t, b_{t-1}) = Z'(s_t, b_{t-1}) \underset{s_{t+1}}{\operatorname{softmin}} Q\left(s_t, \pi^*(s_t, b_{t-1}), \frac{b_{t-1}}{Z'(s_t, b_{t-1})}, s_{t+1}\right), \quad (11)$$

$$Q_R(s_t, a_t, b_t) = \sum_{s_{t+1}} \tau^*(s_{t+1}|s_t, a_t, b_t)\left(R(s_t, a_t, s_{t+1}) + Q_R\left(s_{t+1}, \pi^*(s_{t+1}, b_t), \frac{b_t}{Z'(s_{t+1}, b_t)}\right)\right) \quad (12)$$

*where $Z'(s_t, b_{t-1}) = Z(s_t, \pi^*(s_t, b_{t-1}), b_{t-1})$.*

The proof is given in Appendix A.

Since we have a maximum causal entropy estimation problem, $\tau^*$ (Equation 9) takes the form of a Boltzmann distribution with temperature $\lambda^{-1}$ and energy given by $Q(s_t, a_t, b_t, s_{t+1})$. Function $Q$ (Equation 10) specifies the value of a transition from $s_t, a_t, b_t$ to state $s_{t+1}$. Intuitively, it is a sum of (i) the immediate return, which in turn is a mixture of rewards from $R$ and $\tilde{R}$ weighted by the current belief state, and (ii) the value of the next state $s_{t+1}$ given that the current belief is $b_t$. We have the additional complication that $\pi$ is chosen dynamically as the maximizer of $\rho_R(\pi, \tau)$ rather than statically. Given $\tau^*$, the optimal policy $\pi^*$ (Equation 9) aims at maximizing the expected future return from $R$ defined in (12). Notice that since the optimal policy $\pi^*$ is deterministic and $\tilde{\pi}$ is Markovian, the belief state update rule of (8) can be written in the more concise form: $b_{t+1} = \frac{b_t}{Z'(s_{t+1}, b_t)}$. Finally, given $\tau^*$ and $\pi^*$, we can compute the optimal value $V$ obtained from state $s_t$ and belief state $b_{t-1}$ as defined in (11). Algorithm 1 summarizes our Markovian dynamic program.

In contrast to typical value iteration in discrete MDPs, the belief states are continuous variables in Algorithm 1. In practice, we discretize them by considering a set $\mathcal{B}$ of values in the range $[0, 1]$ and then interpolate between these points. Notice that since $\pi^*$ is deterministic, values in $(0, 0.5)$ are not possible and can be safely neglected. This discretization allows for a compact tabular representation of all functions defined in Theorem 2. The asymptotic complexity of this procedure (Algorithm 1) is then $\mathcal{O}(|\mathcal{S}|^2|\mathcal{A}| \, |\mathcal{B}| \, T)$.

The robust policy $\pi^*$ returned by Algorithm 1 is, for each time-step $t$, a function $\pi_t^* : \mathcal{S} \times \mathcal{B} \to \mathcal{A}$ mapping state-belief

---
**Algorithm 1** Min-max Dynamic Programming
---
**Require:** Reference policy $\tilde{\pi}$, reward function $R(s_t, a_t, s_{t+1})$, feature function $\phi(s_t, a_t, s_{t+1})$, Lagrange multiplier $\omega$, entropy regularization weight $\lambda$
**Ensure:** Robust dynamics $\tau^*$, optimal policy $\pi^*$

---
$V(s_T, b_{T-1}) \leftarrow 0$; $\widetilde{R}(s_t, a_t, s_{t+1}) \leftarrow \omega \cdot \phi(s_t, a_t, s_{t+1})$
**for** $t = T - 1$ to 1 **do**
    Set $Q(s_t, a_t, b_t, s_{t+1})$ from $V$ using (10)
    Set $\tau^*(\cdot|s_t, a_t, b_t) \propto e^{-\lambda Q(s_t, a_t, b_t, \cdot)}$
    Set $Q_R(s_t, a_t, b_t)$ from $\tau^*$ and $Q_R$ using (12)
    Set $\pi^*(s_t, b_{t-1}) = \operatorname{argmax}_{a_t} Q_R(s_t, a_t, b_t)$
    Set $V(s_t, b_{t-1})$ from $Q$ and $\pi^*$ using (11)
**end for**
---

state couples to actions. For the sake of completeness, we show how such a policy can be used in a regular MDP with dynamics $\tau$. Notice that, since belief states are updated according to Equation (8), we need to keep track of the reference policy $\tilde{\pi}$. At the first time-step, state $s_1$ is drawn from the MDP's initial state distribution, while the initial belief state $b_0$ is set to 0.5, as can be seen from Equation (7). Then, action $a_1 = \pi_1^*(s_1, b_0)$ is taken, and the system transitions to the next state $s_2 \sim \tau(\cdot|s_1, a_1)$. Finally, the belief state is updated to account for the choice of action $a_1$: $b_1 = b_0 / N(s_1, b_0)$. Then, this process is repeated until the maximum time-step is reached.

### 3.4 Parameter Optimization

Standard gradient-based methods can be used to optimize the choice of model parameters $\boldsymbol{\omega}$, since the unconstrained dual objective function is a concave function of $\boldsymbol{\omega}$. Any such method is required to repeatedly solve the inner minimax problem of Equation (6) as specified in the previous section, obtaining $(\pi^*, \tau^*)$, compute the feature expectations of the reference policy $\widetilde{\pi}$ under $\tau^*$, and use these to update $\boldsymbol{\omega}$. Conceptually, model parameters $\boldsymbol{\omega}$ are chosen to motivate the adversary's dynamics to satisfy the constraints from the reference policy—(approximately) matching the state-action feature statistics of the training trajectories. Hence, under the assumption that matching features is feasible, following the gradient update rule, $\boldsymbol{\omega}_{i+1} \leftarrow \boldsymbol{\omega}_i + \eta_i(\boldsymbol{\kappa}_\phi(\widetilde{\pi}, \tau^*) - \widehat{\boldsymbol{\kappa}})$, converges when the statistics match, i.e., when $\boldsymbol{\kappa}_\phi(\widetilde{\pi}, \tau^*) = \widehat{\boldsymbol{\kappa}}^3$.

Computing the expected features under the adversary's non-Markovian dynamics, $\tau^*$, requires an extension of the dynamic programming algorithm used to obtain $\tau^*$ itself. The next result follows almost straightforwardly from Theorem 2. For the sake of completeness, we include a proof in Appendix A.

**Corollary 1.** *Let $(\pi^*, \tau^*)$ be the belief-augmented solution of Theorem 2, $p(s_1)$ be the initial state distribution of the given MDP, and $\widetilde{\pi}$ be a randomized Markovian policy. Then:*

$$\boldsymbol{\kappa}_\phi(\widetilde{\pi}, \tau^*) = \sum_{s_1} p(s_1)\boldsymbol{\Psi}(s_1, b_0), \tag{13}$$

*where $\boldsymbol{\Psi}$ is defined recursively for $t = 1, \ldots, T-1$ as:*

$$\boldsymbol{\Psi}(s_t, b_{t-1}) = \sum_{a_t} \widetilde{\pi}(a_t|s_t) \sum_{s_{t+1}} \tau^*(s_{t+1}|s_t, a_t, b_t) \left[\phi(s_t, a_t, s_{t+1}) + \boldsymbol{\Psi}(s_{t+1}, b_t)\right], \tag{14}$$

*with $\boldsymbol{\Psi}(s_T, b_{T-1}) = \mathbf{0}$ and $b_t = \frac{b_{t-1}}{Z(s_t, a_t, b_{t-1})} \mathbb{1}\left[a_t = \pi^*(s_t, b_{t-1})\right]$.*

Notice that the computation of $\boldsymbol{\kappa}_\phi(\widetilde{\pi}, \tau^*)$, as given by Corollary 1, can be efficiently included in the dynamic program of Algorithm 1 by updating $\boldsymbol{\Psi}$ as the last step of each iteration according to (14).

## 4 Experiments

In this section, we empirically evaluate our robust approach for control using uncertainty sets defined by marginal state-action statistics. We consider two different experiments. The first one is a classic grid navigation problem and the second one is a more challenging domain in which the goal is to control the population change of an invasive species. In all experiments, we compare our marginally-constrained approach (MC) to three other methods for estimating the state-transition dynamics: (1) a supervised approach using logistic regression (LR); (2) a robust MDP with *s,a*-rectangular uncertainty sets (RECT); and (3) a simple maximum likelihood estimation (MLE) of the conditional transition probabilities for all state-action pairs. Furthermore, due to the similarity between our settings and batch reinforcement learning, we also compare to fitted Q-iteration (FQI) [29].

### 4.1 Gridworld

We consider an agent navigating through an $N \times N$ grid in order to reach a goal position. The agent's location is described by its horizontal and vertical coordinates $(x, y)$. At each time-step, the agent can attempt to move in each of the four cardinal directions. With probability $p = 0.3$, the action fails and the agent moves in a random direction instead. Attempts to move off the grid have no effect. The agent's initial position is $(1, 1)$, while the goal is to reach state $(N, N)$. The horizon is set to $T = 2N$, while the reward function is the negative $l_1$ distance between the next state and the goal.

In this experiment, we prove the generalization capabilities of our approach. We consider a sequence of gridworlds with increasing size. For each of them, we collect 50 trajectories under a uniform reference policy and we run all algorithms on such data. Intuitively, for small grids, such trajectories provide enough exploration to allow all methods to accurately approximate the state-transition

---

³When $l_1$ or $l_2^2$ regularization of $\boldsymbol{\omega}$ is used, this procedure converges when the feature expectations are close to the sample statistics, where the closeness depends on the amount of regularization used (see Section 3.2).

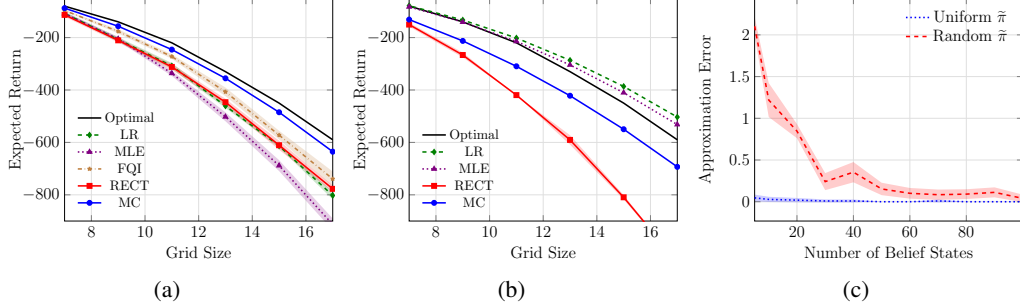

Figure 1: Results of the gridworld experiments, each with 95% confidence intervals. (a) Expected return under the true dynamics as a function of the grid size. (b) Expected return under the estimated (robust) dynamics as function of the grid size. (c) Approximation error incurred by our algorithm due to the discretization of the belief space.

dynamics. However, as the grid grows larger, only a small portion of the state-space is observed in the training data. Thus, generalization is required to achieve good performance. Additional details on the adopted parameters are given in Appendix C.1.

Figure 1a shows the expected return achieved by all algorithms as a function of the grid size $N$. Results are averaged over 20 runs. As expected, for small grids (e.g., $N \leq 7$) all approaches obtain nearly-optimal performance. However, as the grid size increases, only our method is able to estimate dynamics that generalize across unseen regions of the state-space, thus maintaining nearly-optimal performance. FQI is also able to generalize and achieves a significant improvement over the other alternatives, but is not able to compete with our method due to the small number of trajectories available. LR is likely to estimate very optimistic dynamics, thus leading to worse performance. Finally, RECT obtains results comparable to LR even without generalizing. However, rectangular uncertainty sets are too conservative to compete with our method. To better demonstrate this fact, Figure 1b shows the performance achieved by the optimal policy computed by each algorithm under their own estimated dynamics (except for FQI, which is model-free). We clearly notice that the worst-case expected return obtained by the rectangular solution is, as claimed, very conservative. Our approach, on the other hand, shows robust performance comparable to the true ones of the other methods. Due to their optimistic estimates, both LR and MLE obtain an expected return even larger than the optimal one.

Finally, we analyze the approximation error incurred from discretizing the belief states in our approach. We consider a $5 \times 5$ gridworld with the same parameters as before and run the dynamic program of Algorithm 1 for 50 random values of $\boldsymbol{w}$ using two different reference policies: the uniform one and a random one. Figure 1c shows the average absolute deviation of the objective function from its true value as a function of the number of discrete belief states $N_b$. Since, as we can observe from (7), the total number of belief states that are reachable in a finite horizon depends on the number of different probability values assigned by $\widetilde{\pi}$, the uniform reference policy achieves a very small approximation error even with few belief states. Interestingly, the approximation error for a random reference policy, which can be regarded as a 'worst-case' scenario, can also be reduced using a relatively small number of belief states.

## 4.2 Invasive Species

We next consider modeling the population change of an invasive species in an ecosystem with a single action available for mitigating its spread (e.g., introducing a predator). Our starting point is a state-space model with exponential dynamics adapted from Chapter 5 of [30]. Each state captures the current abundance of the invasive species, which we denote as $N_t$ at time $t$. The population evolves according to exponential dynamics, so that $N_{t+1} = \min\{\nu_t N_t, K\}$, where $K$ is the maximum carrying capacity. The growth rate $\nu$ depends on (i) whether the control action $a_t$ has been applied, (ii) the current population level $N_t$, and (iii) random noise. When the control action is not applied ($a_t = 0$), the growth rate is: $\nu_t = \max\{0, \bar{\nu} + \mathcal{N}(0, \sigma_\nu^2)\}$, where $\bar{\nu}$ is the mean growth rate. In this case, the growth rate is independent of the current population level. When the control action is applied ($a_t = 1$), the growth rate is: $\nu_t = \bar{\nu} - \beta_1 N_t - \beta_2 \max\{0, N_t - \hat{N}\}^2 + \mathcal{N}(0, \sigma_\nu^2)$, where

Table 1: Negative expected return for different numbers of trajectories $M$ and reference policy's control probabilities $p$ in the invasive species experiment. Each value is the average of 20 independent runs. 95% confidence intervals are shown. The best algorithms are highlighted in bold.

| Alg. | $M$ | $p = 0.1$ | $p = 0.2$ | $p = 0.3$ | $p = 0.4$ | $p = 0.5$ |
|------|-----|-----------|-----------|-----------|-----------|-----------|
| MLE | 50 | $121.74 \pm 0.82$ | $128.34 \pm 2.06$ | $140.36 \pm 1.28$ | $147.189 \pm 1.78$ | $149.82 \pm 2.12$ |
| LR | 50 | $152.95 \pm 13.5$ | $106.77 \pm 2.21$ | $117.43 \pm 5.09$ | $122.756 \pm 5.94$ | $\mathbf{123.28} \pm 4.82$ |
| MC | 50 | $\mathbf{99.37} \pm 0.96$ | $\mathbf{102.38} \pm 1.82$ | $\mathbf{98.36} \pm 0.78$ | $\mathbf{107.39} \pm 3.44$ | $\mathbf{124.47} \pm 1.81$ |
| RECT | 50 | $111.91 \pm 5.33$ | $\mathbf{107.71} \pm 4.13$ | $117.15 \pm 6.76$ | $123.55 \pm 7.95$ | $142.26 \pm 8.28$ |
| FQI | 50 | $140.85 \pm 6.11$ | $133.08 \pm 5.36$ | $133.77 \pm 4.70$ | $134.05 \pm 6.22$ | $140.25 \pm 5.04$ |
| MLE | 100 | $120.91 \pm 0.63$ | $125.21 \pm 1.25$ | $134.23 \pm 1.33$ | $140.96 \pm 1.76$ | $145.42 \pm 1.72$ |
| LR | 100 | $169.27 \pm 8.72$ | $\mathbf{104.70} \pm 3.43$ | $110.09 \pm 2.57$ | $114.23 \pm 2.49$ | $\mathbf{124.53} \pm 4.98$ |
| MC | 100 | $\mathbf{98.25} \pm 0.88$ | $103.66 \pm 1.05$ | $\mathbf{96.20} \pm 0.95$ | $105.17 \pm 1.95$ | $115.04 \pm 6.18$ |
| RECT | 100 | $\mathbf{100.98} \pm 3.33$ | $103.80 \pm 3.22$ | $108.69 \pm 4.95$ | $\mathbf{106.18} \pm 4.02$ | $136.24 \pm 8.41$ |
| FQI | 100 | $126.66 \pm 5.84$ | $121.93 \pm 6.27$ | $119.85 \pm 4.30$ | $125.65 \pm 5.08$ | $131.51 \pm 4.92$ |

$\beta_1$ and $\beta_2$ are the coefficients of effectiveness and $\hat{N}$ is the population at which the effectiveness peaks. That is, the effectiveness of the control method may increase or decrease depending on the population of the invasive species. This dependence is modeled using a simplified quadratic spline. The precise population $N_t$ of the species cannot be directly observed. Instead, one can observe a noisy estimate $y_t = N_t + \mathcal{N}(0, \sigma_y^2)$. The exact values of the parameters used in this experiment are $K = 500$, $T = 100$, $\hat{K} = 300$, $\bar{\nu} = 1.02$, $\beta_1 = 0.001$, $\beta_2 = -0.0000021$, $\sigma_\nu^2 = 0.02$, $\sigma_y^2 = 20$. Notice that due to its highly unstable dynamics and noisy observations, this domain represents a very challenging control problem.

In this experiment, we analyze the behavior of all algorithms when given different amounts of trajectories collected under different reference policies. In particular, we consider five reference policies, where each chooses to apply the control action with a fixed probability $p \in \{0.1, 0.2, 0.3, 0.4, 0.5\}$. For each reference policy, we generate two datasets of $M_1 = 50$ trajectories and $M_2 = 100$ trajectories, respectively. Additional details are given in Appendix C.2.

Results of our experiments in these settings are reported in Table 1. Each datapoint is obtained as the result of an average over 20 runs, We notice that MC outperforms all alternatives when $p < 0.5$ and $M = 50$. As before, this is due to its generalization capabilities. When considering $M = 100$ trajectories, all other approaches significantly improve their performance. However, MC is still able to achieve better results for most values of $p$. The rectangular solution (RECT) also achieves good performance, but shows a much higher variability. Finally, we note that all algorithms suffer from the very limited exploration provided by a reference policy with $p = 0.5$. In such cases, the performance of the feature-based approaches are superior.

## 5 Conclusion & Future Work

In this paper, we have proposed a new approach to robust control based on causally conditioned probability distribution estimation that defines uncertainty sets using features of the interaction with the decision process with a different policy. Though the solution to the corresponding robust control problem is non-Markovian, we show that it can be closely approximated by augmenting the typical Markovian robust MDP formulation [31, 5] with a continuous-valued "belief state" that can then be discretized. We have empirically tested our approach on a synthetic experiment and a real-world control problem, highlighting its advantages over methods that form rectangular uncertainty sets.

We plan to extend our formulation to incorporate constraints that are obtained from multiple separate reference control policies. This could also allow episodic reinforcement learning [32] where the robust optimal control policy is employed and then updated based on the trajectories that are observed from its application. Incorporating more sophisticated ideas for solving POMDPs using belief state compression will likely be required, since discretizing the belief space scales poorly with the number of different reference policies.

**Acknowledgments**

We thank the anonymous reviewers whose comments helped to improve the paper significantly. This work was supported, in part, by the National Science Foundation under Grant No. 1652530 and Grant No. 1717368, and by the Future of Life Institute (futureoflife.org) FLI-RFP-AI1 program.

## Footnotes

[1]Notice that $\tau$ must also belong to the set of valid probability distributions. We omit the corresponding constraints for the sake of clarity.

[2]Without any loss of generality, this problem could be equivalently posed as finding the control policy $\pi$ that maximizes a mixture of rewards $\theta \rho_R(\pi, \tau) + (1-\theta)\rho_{\tilde{R}}(\pi, \tilde{\tau})$ for two different decision processes with dynamics/reward $(\tau, R)$ and $(\tilde{\tau}, \tilde{R})$.

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
