[Supplementary Material · supplementary.pdf]

# A Proofs

## A.1 Proof of Theorem 1

*Proof of Theorem 1.* Lagrangianizing the constraints of the optimization problem of Definition 2, we obtain:

$$\max_{\pi} \min_{\tau} \max_{\boldsymbol{\omega}} \left( \rho(\pi,\tau) - \frac{1}{\lambda} H_{\tau,\bar{\pi}}(S_{1:T}||A_{1:T-1}) + \boldsymbol{\omega} \cdot (\boldsymbol{\kappa}_{\phi}(\widetilde{\pi},\tau) - \widehat{\boldsymbol{\kappa}}) \right).$$

Under the assumption that there exists $\tau$ that matches the features while strictly satisfying the probabilistic inequality constraints (which we kept implicit for the sake of clarity), Slater's condition and, thus, strong duality, holds [33]. Then, we can write:

$$\max_{\boldsymbol{\omega}} \max_{\pi} \min_{\tau} \left( \rho(\pi,\tau) + \boldsymbol{\omega} \cdot \boldsymbol{\kappa}_{\phi}(\widetilde{\pi},\tau) - \frac{1}{\lambda} H_{\tau,\bar{\pi}}(S_{1:T}||A_{1:T-1}) \right) - \boldsymbol{\omega} \cdot \widehat{\boldsymbol{\kappa}}.$$

The inner minimization problem in this expression is equivalent to a maximum causal entropy estimation problem scaled by $\lambda$ [21] and has a *softmin* Boltzmann distribution over state outcomes,

$$\operatorname*{softmin}_{x} f(x) = -\frac{1}{\lambda} \log \sum_{x} e^{-\lambda f(x)},$$

which is a smooth relaxation of the *min* function. Thus, we can safely remove the entropy term, while replacing *min* with *softmin*:

$$\max_{\boldsymbol{\omega}} \max_{\pi} \operatorname*{softmin}_{\tau} \left( \rho(\pi,\tau) + \boldsymbol{\omega} \cdot \boldsymbol{\kappa}_{\phi}(\widetilde{\pi},\tau) \right) - \boldsymbol{\omega} \cdot \widehat{\boldsymbol{\kappa}}.$$

□

## A.2 Proof of Theorem 2

*Proof of Theorem 2.* For the sake of completeness, we derive the theorem for the zero-sum game where the entropy regularizer is explicitly stated, although we already know from Theorem 1 that it will lead to a softmin solution. Thus, we want to solve the problem:

$$\max_{\pi} \min_{\tau} \left\{ \rho_R(\pi,\tau) + \rho_{\widetilde{R}}(\widetilde{\pi},\tau) - \frac{1}{\lambda} H_{\tau,\bar{\pi}}(S_{1:T}||A_{1:T-1}) \right\}.$$

Consider the computation of the distribution over final states $s_T$ after state-action sequence $\mathbf{s}_{1:T-1}$, $\mathbf{a}_{1:T-1}$, namely $\tau(\cdot|\mathbf{s}_{1:T-1}, \mathbf{a}_{1:T-1})$. Specifically, we want to solve the optimization problem:

$$\min_{\tau(\cdot|\mathbf{s}_{1:T-1}, \mathbf{a}_{1:T-1})} L(\tau, \mathbf{s}_{1:T-1}, \mathbf{a}_{1:T-1}),$$

where:

$$L(\tau, \mathbf{s}_{1:T-1}, \mathbf{a}_{1:T-1}) = \tau(\mathbf{s}_{1:T-1}||\mathbf{a}_{1:T-2}) \sum_{s_T} \tau(s_T|\mathbf{s}_{1:T-1}, \mathbf{a}_{1:T-1}) \qquad (15)$$

$$\left( \pi(\mathbf{a}_{1:T-1}||\mathbf{s}_{1:T-1})R(s_{T-1}, a_{T-1}, s_T) \right.$$

$$+ \ \widetilde{\pi}(\mathbf{a}_{1:T-1}|\mathbf{s}_{1:T-1})\widetilde{R}(s_{T-1}, a_{T-1}, s_T)$$

$$+ \ \frac{1}{\lambda} \bar{\pi}(\mathbf{a}_{1:T-1}|\mathbf{s}_{1:T-1}) \log \tau \left( s_T|\mathbf{s}_{1:T-1}, \mathbf{a}_{1:T-1} \right) \Big).$$

Notice that we kept implicit the positivity constraints, $\tau(s_T|\mathbf{s}_{1:T-1}, \mathbf{a}_{1:T-1}) >= 0$, and the normalization ones, $\sum_{s_T} \tau(s_T|\mathbf{s}_{1:T-1}, \mathbf{a}_{1:T-1}) = 1$. Differentiating with respect to $\tau(s_T|\mathbf{s}_{1:T-1}, \mathbf{a}_{1:T-1})$ we obtain:

$$\frac{\partial L}{\partial \tau} = \tau(\mathbf{s}_{1:T-1}||\mathbf{a}_{1:T-2}) \Big( (\pi(\mathbf{a}_{1:T-1}||\mathbf{s}_{1:T-1})R(s_{T-1}, a_{T-1}, s_T)$$

$$+ \ \widetilde{\pi}(\mathbf{a}_{1:T-1}|\mathbf{s}_{1:T-1})\widetilde{R}(s_{T-1}, a_{T-1}, s_T)$$

$$+ \ \lambda^{-1} \bar{\pi}(\mathbf{a}_{1:T-1}|\mathbf{s}_{1:T-1}) \log \tau(s_T|\mathbf{s}_{1:T-1}, \mathbf{a}_{1:T-1})$$

$$+ \ \lambda^{-1} \bar{\pi}(\mathbf{a}_{1:T-1}|\mathbf{s}_{1:T-1}) \Big).$$

Equating this last term to zero and solving for $\tau(s_T|\mathbf{s}_{1:T-1}, \mathbf{a}_{1:T-1})$, we obtain:

$$\tau(s_T|\mathbf{s}_{1:T-1}, \mathbf{a}_{1:T-1}) \propto e^{-\lambda Q(\mathbf{s}_{1:T-1}, \mathbf{a}_{1:T-1}, s_T)}$$

where we set:

$$Q(\mathbf{s}_{1:T-1}, \mathbf{a}_{1:T-1}, s_T) = \frac{\pi(\mathbf{a}_{1:T-1}||\mathbf{s}_{1:T-1})R(s_{T-1}, a_{T-1}, s_T) + \widetilde{\pi}(\mathbf{a}_{1:T-1}|\mathbf{s}_{1:T-1})\widetilde{R}(s_{T-1}, a_{T-1}, s_T)}{\bar{\pi}(\mathbf{a}_{1:T-1}|\mathbf{s}_{1:T-1})}$$

Notice that we neglect the term $\lambda\bar{\pi}(\mathbf{a}_{1:T-1}|\mathbf{s}_{1:T-1})$ since it is constant over next states. Although convenient, this form requires the knowledge of the whole state-action history in order to compute $\tau$. However, since $\bar{\pi}$ is arbitrary and do not affect our solution, we can set it to $(\pi + \tilde{\pi})/2$, thus obtaining:

$$Q(\mathbf{s}_{1:T-1}, \mathbf{a}_{1:T-1}, s_T) = \frac{\pi(\mathbf{a}_{1:T-1}||\mathbf{s}_{1:T-1})R(s_{T-1}, a_{T-1}, s_T) + \widetilde{\pi}(\mathbf{a}_{1:T-1}|\mathbf{s}_{1:T-1})\widetilde{R}(s_{T-1}, a_{T-1}, s_T)}{\pi(\mathbf{a}_{1:T-1}||\mathbf{s}_{1:T-1}) + \widetilde{\pi}(\mathbf{a}_{1:T-1}|\mathbf{s}_{1:T-1})}$$

$$= b_{T-1}R(s_{T-1}, a_{T-1}, s_T) + (1 - b_{T-1})\widetilde{R}(s_{T-1}, a_{T-1}, s_T),$$

where we neglect the constant term $2$ since it can easily be embedded into $\lambda$. Here, we introduce a continuous "belief state" summarizing the history of states and actions:

$$b_t \triangleq \frac{\pi(\mathbf{a}_{1:t}||\mathbf{s}_{1:t})}{\pi(\mathbf{a}_{1:t}||\mathbf{s}_{1:t}) + \widetilde{\pi}(\mathbf{a}_{1:t}||\mathbf{s}_{1:t})}.$$

Notice that now $Q(s_{T-1}, a_{T-1}, b_{T-1}, s_T)$ depends only on variables at time $T-1$, and so does $\tau$. Then, our final next-state distribution is obtained after normalization:

$$\tau^*(s_T|s_{T-1}, a_{T-1}, b_{T-1}) = \frac{e^{-\lambda Q(s_{T-1}, a_{T-1}, b_{T-1}, s_T)}}{\sum_{s'_T} e^{-\lambda Q(s_{T-1}, a_{T-1}, b_{T-1}, s'_T)}}. \tag{16}$$

Continuing backwards, the optimal action for $\pi$ to take in state $s_{T-1}$ and belief state $b_{T-2}$ is:

$$\pi^*(s_{T-1}, b_{T-2}) = \underset{a_{T-1}}{\operatorname{argmax}} Q_R\left(s_{T-1}, a_{T-1}, \frac{b_{T-2}}{Z(s_{T-1}, a_{T-1}, b_{T-2})}\right), \tag{17}$$

where we define $Z(s_t, a_t, b_{t-1}) \triangleq b_{t-1} + (1 - b_{t-1})\tilde{\pi}(a_t|s_t)$ and $Q_R$ is the expected reward under $\tau^*$:

$$Q_R(s_{T-1}, a_{T-1}, b_{T-1}) = \sum_{s_T} \tau^*(s_T|s_{T-1}, a_{T-1}, b_{T-1})R(s_{T-1}, a_{T-1}, s_T).$$

Equation (17) can be verified by noticing that, after taking an action $a_{T-1}$, the belief state $b_{T-2}$ must be updated according to (8). Furthermore, it is easy to verify that the optimal policy is deterministic by writing the objective as a function of the whole history rather than belief states.

Given the distribution over final states $s_T$ and the final action $a_{T-1}$, we can compute the objective value at time $T-1$ by substituting Equation (16) and 17 into Equation (15):

$$V(s_{T-1}, b_{T-2}) = \sum_{s_T} \tau^*\left(s_T|s_{T-1}, \pi^*(s_{T-1}, b_{T-2}), \frac{b_{T-2}}{Z'(s_{T-1}, b_{T-2})}\right)$$

$$\left(\pi(\mathbf{a}_{1:T-1}||\mathbf{s}_{1:T-1})R(s_{T-1}, \pi^*(s_{T-1}, b_{T-2}), s_T)\right.$$

$$+ \ \widetilde{\pi}(\mathbf{a}_{1:T-1}|\mathbf{s}_{1:T-1})\widetilde{R}(s_{T-1}, \pi^*(s_{T-1}, b_{T-2}), s_T)$$

$$- \ \bar{\pi}(\mathbf{a}_{1:T-1}|\mathbf{s}_{1:T-1})Q\left(s_{T-1}, \pi^*(s_{T-1}, b_{T-2}), \frac{b_{T-2}}{Z'(s_{T-1}, b_{T-2})}, s_T\right)$$

$$\left. - \ \lambda^{-1}\bar{\pi}(\mathbf{a}_{1:T-1}|\mathbf{s}_{1:T-1})\log\sum_{s'_T} e^{-\lambda Q\left(s_{T-1}, \pi^*(s_{T-1}, b_{T-2}), \frac{b_{T-2}}{Z'(s_{T-1}, b_{T-2})}, s'_T\right)}\right)$$

$$= \ -\lambda^{-1}\bar{\pi}(\mathbf{a}_{1:T-1}|\mathbf{s}_{1:T-1})\log\sum_{s'_T} e^{-\lambda Q\left(s_{T-1}, \pi^*(s_{T-1}, b_{T-2}), \frac{b_{T-2}}{Z'(s_{T-1}, b_{T-2})}, s'_T\right)}$$

$$= \ Z'(s_{T-1}, b_{T-2})\underset{s_T}{\operatorname{softmin}} Q\left(s_{T-1}, \pi^*(s_{T-1}, b_{T-2}), \frac{b_{T-2}}{Z'(s_{T-1}, b_{T-2})}, s_T\right),$$

where we defined $Z'(s_t, b_{t-1}) \triangleq b_{t-1} + (1 - b_{t-1})\tilde{\pi}(\pi^*(s_t, b_{t-1})|s_t) = Z(s_t, \pi^*(s_t, b_{t-1}), b_{t-1})$.

We can now move to the preceding time-step to choose $\tau(\cdot|s_{T-2}, a_{T-2}, b_{T-2})$. Similarly as before, we want to compute:

$$\min_{\tau(\cdot|s_{T-2}, a_{T-2}, b_{T-2})} L(\tau, s_{T-2}, a_{T-2}, b_{T-3}),$$

where we replace the dependency on the full history with the previously-defined belief state:

$$L(\tau, s_{T-2}, a_{T-2}, b_{T-3}) = \sum_{s_{T-1}} \tau(s_{T-1}|s_{T-2}, a_{T-2}, b_{T-2}) Z(s_{T-2}, a_{T-2}, b_{T-3})$$

$$\Big( b_{T-2} R(s_{T-2}, a_{T-2}, s_{T-1})$$
$$+ (1 - b_{T-2})\tilde{R}(s_{T-2}, a_{T-2}, s_{T-1})$$
$$+ V(s_{T-1}, b_{T-2})$$
$$+ \lambda^{-1} \log \tau(s_{T-1}|s_{T-2}, a_{T-2}, b_{T-2})\Big).$$

By differentiating, equating to zero, and solving for $\tau$, we obtain:

$$\tau^*(s_{T-1}|s_{T-2}, a_{T-2}, b_{T-2}) = \frac{e^{-\lambda Q(s_{T-2}, a_{T-2}, b_{T-2}, s_{T-1})}}{\sum_{s'_{T-1}} e^{-\lambda Q(s_{T-2}, a_{T-2}, b_{T-2}, s'_{T-1})}},$$

where:

$$Q(s_{T-2}, a_{T-2}, b_{T-2}, s_{T-1}) = b_{T-2} R(s_{T-2}, a_{T-2}, s_{T-1})$$
$$+ (1 - b_{T-2})\tilde{R}(s_{T-2}, a_{T-2}, s_{T-1})$$
$$+ V(s_{T-1}, b_{T-2}).$$

Similarly as before, we can compute the optimal action $a_{T-2}$ as:

$$\pi^*(s_{T-2}, b_{T-3}) = \operatorname*{argmax}_{a_{T-2}} Q_R\left(s_{T-2}, a_{T-2}, \frac{b_{T-3}}{Z(s_{T-2}, a_{T-2}, b_{T-3})}\right),$$

where $Q_R$ is, similarly to $Q$, augmented with the future expectation:

$$Q_R(s_{T-2}, a_{T-2}, b_{T-2}) = \sum_{s_{T-1}} \tau^*(s_{T-1}|s_{T-2}, a_{T-2}, b_{T-2})\Big(R(s_{T-2}, a_{T-2}, s_{T-1})$$

$$+ Q_R(s_{T-1}, \pi^*(s_{T-1}, b_{T-2}), \frac{b_{T-2}}{Z'(s_{T-1}, b_{T-2})})\Big).$$

Finally, the objective value is:

$$V(s_{T-2}, b_{T-3}) = \sum_{s_{T-1}} \tau^*\left(s_{T-1}|s_{T-2}, \pi^*(s_{T-2}, b_{T-3}), \frac{b_{T-3}}{Z'(s_{T-2}, b_{T-3})}\right) Z'(s_{T-2}, b_{T-3})$$

$$\Bigg(\frac{b_{T-3}}{Z'(s_{T-2}, b_{T-3})} R(s_{T-2}, \pi^*(s_{T-2}, b_{T-3}), s_{T-1})$$

$$+ \left(1 - \frac{b_{T-3}}{Z'(s_{T-2}, b_{T-3})}\right) \tilde{R}(s_{T-2}, \pi^*(s_{T-2}, b_{T-3}), s_{T-1})$$

$$+ V\left(s_{T-1}, \frac{b_{T-3}}{Z'(s_{T-2}, b_{T-3})}\right)$$

$$- Q\left(s_{T-2}, \pi^*(s_{T-2}, b_{T-3}), \frac{b_{T-3}}{Z'(s_{T-2}, b_{T-3})}, s_{T-1}\right)$$

$$- \lambda^{-1} \log \sum_{s'_{T-1}} e^{-\lambda Q\left(s_{T-2}, \pi^*(s_{T-2}, b_{T-3}), \frac{b_{T-3}}{Z'(s_{T-2}, b_{T-3})}, s'_{T-1}\right)}\Bigg)$$

$$= Z'(s_{T-2}, b_{T-3}) \operatorname*{softmin}_{s_{T-1}} Q\left(s_{T-2}, \pi^*(s_{T-2}, b_{T-3}), \frac{b_{T-3}}{Z'(s_{T-2}, b_{T-3})}, s_{T-1}\right).$$

Continuing this procedure backwards, alternating the computation of transitions and optimal actions, completes the proof. $\qquad\square$

## A.3 Proof of Corollary 1

*Proof of Corollary 1.* We start by showing that:

$$\mathbf{\Psi}(s_t, b_{t-1}) = \mathbb{E}_{\widetilde{\pi}, \tau^*} \left[ \sum_{i=t}^{T-1} \phi(s_i, a_i, s_{i+1}) | s_t, b_{t-1} \right]. \tag{18}$$

This can be easily proven by induction using similar steps as in the proof of Theorem 2. Clearly, at time step $T - 1$:

$$\mathbf{\Psi}(s_{T-1}, b_{T-2}) = \sum_{a_{T-1}} \widetilde{\pi}(a_{T-1}|s_{T-1}) \sum_{s_T} \tau^*(s_T|s_{T-1}, a_{T-1}, b_{T-1}) \phi(s_{T-1}, a_{T-1}, s_T).$$

Recalling the belief update rule (8), we have:

$$b_{T-1} = \frac{b_{T-2} \mathbb{1} \left[ a_{T-1} = \pi^*(s_{T-1}, b_{T-2}) \right]}{b_{T-2} \mathbb{1} \left[ a_{T-1} = \pi^*(s_{T-1}, b_{T-2}) \right] + (1 - b_{T-2}) \widetilde{\pi}(a_{T-1}|s_{T-1})},$$

where the indicator is due to the fact that $\pi^*$ is deterministic. Notice that this is equivalent to the more concise update rule given in the main statement. Thus, (18) holds at time $T - 1$. Assume now that (18) holds at time $t + 1$. Let us show that this implies (18) holds at time $t$ as well. We have:

$$\mathbf{\Psi}(s_t, b_{t-1}) = \sum_{a_t} \widetilde{\pi}(a_t|s_t) \sum_{s_{t+1}} \tau^*(s_{t+1}|s_t, a_t, b_t) \left[ \phi(s_t, a_t, s_{t+1}) + \mathbf{\Psi}(s_{t+1}, b_t) \right]$$

$$= \sum_{a_t} \widetilde{\pi}(a_t|s_t) \sum_{s_{t+1}} \tau^*(s_{t+1}|s_t, a_t, b_t) \left( \phi(s_t, a_t, s_{t+1}) + \mathbb{E}_{\widetilde{\pi}, \tau^*} \left[ \sum_{i=t+1}^{T-1} \phi(s_i, a_i, s_{i+1}) | s_{t+1}, b_t \right] \right)$$

$$= \mathbb{E}_{\widetilde{\pi}, \tau^*} \left[ \sum_{i=t}^{T-1} \phi(s_i, a_i, s_{i+1}) | s_t, b_{t-1} \right],$$

where the last equation holds since $b_{t-1}$ is again correctly updated as given in the main statement. Finally, since $b_0 = 0.5$ is the only possible initial belief state:

$$\boldsymbol{\kappa}_\phi(\widetilde{\pi}, \tau^*) = \mathbb{E}_{\widetilde{\pi}, \tau^*} \left[ \sum_{t=1}^{T-1} \phi(s_t, a_t, s_{t+1}) \right]$$

$$= \sum_{s_1} p(s_1) \mathbb{E}_{\widetilde{\pi}, \tau^*} \left[ \sum_{t=1}^{T-1} \phi(s_t, a_t, s_{t+1}) | s_1 \right]$$

$$= \sum_{s_1} p(s_1) \mathbf{\Psi}(s_1, b_0).$$

This concludes the proof. $\square$

$$a_1, 1, 1 - \epsilon$$
$$a_2, 1, \epsilon$$

$$a_1, 0, \epsilon \qquad \qquad a_1, 0, \epsilon$$
$$a_2, 0, 1 - \epsilon \qquad s_1 \qquad s_2 \qquad a_2, 0, 1 - \epsilon$$

$$a_1, 1, 1 - \epsilon$$
$$a_2, 1, \epsilon$$

Figure 2: The two-state MDP used throughout this section. The notation $a, r, p$ on the arcs denotes a stochastic transition performed by action $a$ with probability $p$ and reward $r$. The initial state is $s_1$.

## B  A Simple Two-State MDP

In order to provide a better understanding of the proposed approach, we begin by considering a simple example where we can clearly appreciate the effect of all elements involved in our algorithm.

Consider the simple MDP of Figure 2. The MDP has two states ($s_1$ and $s_2$) and two actions ($a_1$ and $a_2$). Action $a_1$ forces the system to change state, while $a_2$ stays in the current state. All actions fail with probability $\epsilon \in [0, 0.5]$ and succeed with probability $1 - \epsilon$. The reward is $1$ whenever the state is changed, $0$ otherwise. The system starts deterministically in state $s_1$ and runs for $T = 4$ time steps. Clearly, the optimal policy is to execute $a_1$ in all states for $\epsilon < 0.5$, while any policy is optimal for $\epsilon = 0.5$.

We now establish some common parameters that will be used throughout this section. Then, we analyze the effect of each of them separately. First, we need to specify features that allow us to define the marginal constraints of Section 3.1. In order to better visualize the results, we adopt only one feature (so that our objective is a function defined on the real line). A useful property we would like to capture is the fact that action $a_1$ frequently changes the current state. Thus, we define our (scalar) feature function as:

$$\phi(s, a, s') = \begin{cases} 1 & \text{if } s \neq s' \text{ and } a = a_1 \\ 0 & \text{otherwise} \end{cases}$$

Notice that the feature expectations $\kappa_\phi(\pi, \tau)$ of some policy $\pi$ under some transition probabilities $\tau$ are equivalent to the expected number of times in which $a_1$ changes state. Thus, if we assume $\pi$ to choose $a_1$ with a fixed probability $p$ (independently of the state) and that the horizon is $T$, we have:

$$\kappa_\phi(\pi, \tau_\epsilon) = p(1 - \epsilon)(T - 1),$$

where $\tau_\epsilon$ denotes the transition probabilities of our two-state MDP with probability of failure $\epsilon$. Unless otherwise stated, we adopt the true feature expectations instead of the sample statistics $\hat{\kappa}_\phi$.

We consider a uniform reference policy $\tilde{\pi}$ (hence $p = 0.5$). Notice that, under uniform $\tilde{\pi}$, the set of all reachable belief states in a fixed horizon $T$ scales linearly with $T$ rather than exponentially[4]:

$$\mathcal{B} = \{0\} \cup \left\{ \frac{0.5}{0.5 + 0.5^t} \mid t = 1, \ldots, T \right\}$$

This fact can be better verified from the belief update rule (8). Thus, in this small domain we can adopt the full belief set so that no approximation error will be incurred. With $T = 4$, we have $\mathcal{B} = \{0, 0.5, 0.67, 0.8, 0.89\}$.

Finally, we use no entropy, $l_1$, or $l_2^2$ regularization.

**The effect of stochasticity**  We start by analyzing how the objective function (6) and the corresponding solution change for different values of $\epsilon$, i.e., for different levels of stochasticity in the underlying system. From Figure 3a, we can notice that the optimal solution lies in an interval when the system is deterministic, while, as we increase the stochasticity through $\epsilon$, it moves on an increasingly peaked corner. Furthermore, the algorithm becomes more conservative for larger $\epsilon$, as can be noticed from the expected return achieved by the min-max solution for different weights

Figure 3: (a) The objective function for different values of $\epsilon$. The higher is the stochasticity, the more peaked is the global optimum. (b) The robust expected return for different values of the Lagrange multiplier $w$. (c) The normalized worst-case performance (solid line) and the true expected return (dashed line) achieved by the optimal solution for different values of $\epsilon$. Here $\pi_\epsilon$ denotes the optimal policy under dynamics $\tau_\epsilon$, while $(\pi^*, \tau^*)$ denote the min-max solution at the corresponding optimum.

Figure 4: (a) Approximation error due to missing belief states, and (b) how it can be alleviated by adding entropy regularization ($\lambda = 20$). (c) The objective function for different entropy regularizers.

$w$ (Figure 3b). This can be better observed from Figure 3c, which shows the true and worst-case expected returns achieved by the global maximizer for each $\epsilon$, normalized by the performance of the corresponding optimal policy. This decrease in the worst-case performance is expected since one feature is not sufficient for constraining the solution with high stochasticity. Interestingly, the performance under the true dynamics $\tau_\epsilon$ remains optimal for all $\epsilon$, which implies that the chosen feature is sufficient for characterizing the optimal policy.

**The effect of misspecified belief states** When a non-uniform reference policy is used, the reachable belief states cannot be efficiently enumerated as we did before. Thus, in practice we approximate them with a smaller set. We now investigate the consequences of such approximation on the objective function. We consider discretizing the belief space uniformly with $0.1$ step, thus obtaining the set $\widetilde{\mathcal{B}} = \{0, 0.5, 0.6, 0.7, 0.8, 0.9\}$. Notice that values in $(0, 0.5)$ cannot occur and are safely removed. Figure 4a shows the resulting approximate objective function (from now on we use $\epsilon = 0.3$). As expected, the two missing belief states, which are now approximated with the closest one in $\widetilde{\mathcal{B}}$, result in two small deviations from the ideal objective. These can be easily alleviated by using a more fine-grained discretization or, as we shall see, by smoothing the objective.

**The effect of entropy regularization** Although subgradient methods are guaranteed to converge under general assumptions, optimizing an almost non-differentiable objective function like the one of Figure 3a can be very slow for high-dimensional problems[5]. Adding a small amount of entropy regularization can dramatically improve the smoothness of the objective function and the corresponding gradient, thus simplifying its optimization. Figure 4c shows an example. Interestingly, the entropy regularizer can also be adopted to alleviate the approximation error due to discretizing the belief space (Figure 4b), which can be much more efficient than increasing the size of $\widetilde{\mathcal{B}}$.

Figure 5: (a) The objective function using $l_1$ and $l_2$ regularization. (b) The objective functions of different reference policies. (c) Expected returns for increasingly good reference policies.

**The effect of estimating feature expectations**  In practice, the feature expectations cannot be computed in closed-form as we did so far but must be estimated from given trajectories. Consequently, $l_1$ or $l_2^2$ regularization of the dual variables is typically required to enforce softened constraints, as the estimation error might make the exact optimization problem of Definition 2 infeasible. Even if the problem remained feasible, a deviation from the ideal feature expectations could void our robustness guarantees since the true transition probabilities might not be part of the uncertainty set anymore. We now show an example demonstrating this fact. Assume that the previously introduced feature expectations $\kappa_\phi$ are now estimated from a small amount of data, obtaining a value $\hat{\kappa}_\phi$ that deviates from the true one by $0.4$, i.e., $\hat{\kappa}_\phi = \kappa_\phi + 0.4$. The solid line in Figure 5a shows the resulting objective (once again, we fix $\epsilon = 0.3$). As we can notice, the optimum has been considerably shifted from its ideal value of Figure 3a. Furthermore, the worst-case expected return achieved at such point, $\rho(\pi^*, \tau^*)$, is now larger than the one achieved by the optimal policy $\rho(\pi_\epsilon, \tau_\epsilon)$, which implies that our solution is not robust anymore. However, we can easily solve this issue by regularizing the Lagrange multiplier $w$. Figure 5a shows the results when adding $l_1$ regularization with parameter $\beta$ (which enforces the soft constraint $|\kappa_\phi - \hat{\kappa}_\phi| \leq \beta$) and $l_2^2$ regularization with parameter $\alpha$. As we can see, for properly chosen parameters, the optimal weight coincides with the original one and so does the worst-case expected return.

**The effect of the reference policy**  So far, we have been using a uniform (non-informative) reference policy. Let us now investigate what happens when the reference policy gets closer to the optimal one. We set $\tilde{\pi}(a_1|s) = p$ and $\tilde{\pi}(a_2|s) = 1 - p$ for all states $s$. Figure 5b shows how the objective function varies when increasing $p$ above $0.5$. Furthermore, Figure 5c shows that the worst-case expected return gets closer to the optimal one as $p$ increases (i.e., as the reference policy becomes optimal). This is again expected since, when the two policies involved in our min-max problem are equivalent, the solution becomes Markovian. Although the single feature we have been using so far is not enough for constraining history-dependent transition dynamics, it suffices for Markovian ones, thus resulting in an optimal worst-case expected return.

## C    Additional Details on the Experiments

### C.1    Gridworld

We provide additional details on the gridworld experiments.

**MC**    In order to define our marginal constraints, we consider 6 features:

$$\phi_1(s, a, s') = \mathbb{1}\left[a = \text{UP} \wedge (s'_y > s_y \wedge s_y < N \vee s'_y = s_y \wedge s_y = N)\right]$$
$$\phi_2(s, a, s') = \mathbb{1}\left[a = \text{RIGHT} \wedge (s'_x > s_x \wedge s_x < N \vee s'_x = s_x \wedge s_x = N)\right]$$
$$\phi_3(s, a, s') = \mathbb{1}\left[a = \text{DOWN} \wedge (s'_y < s_y \wedge s_y > 0 \vee s'_y = s_y \wedge s_y = 0)\right]$$
$$\phi_4(s, a, s') = \mathbb{1}\left[a = \text{LEFT} \wedge (s'_x < s_x \wedge s_x > 0 \vee s'_x = s_x \wedge s_x = 0)\right]$$
$$\phi_5(s, a, s') = \mathbb{1}\left[|s_x - s'_x| + |s_y - s'_y| > 1\right]$$
$$\phi_6(s, a, s') = \mathbb{1}\left[s = s' \wedge s_x \in (1, N) \wedge s_y \in (1, N)\right]$$

Intuitively, the first 4 features are enabled by successful transitions. For instance, feature $\phi_1$ is 1 whenever action UP correctly increases the $y$ coordinate of the current state (or stays at the upper border). The last two features encode deterministic properties of the environment. $\phi_5$ is enabled whenever a transition to a non-adjacent state is performed. This is impossible in the true MDP, so the corresponding feature expectation must be zero. Similarly, $\phi_6$ is enabled whenever an action stays in the current state while not at the borders. This is, again, impossible in the true MDP.

For solving the dynamic program of Algorithm 1, we enumerate all possible belief states reachable in a horizon $T = 6^6$. Notice that, for longer horizons, all belief values are above 0.999 and can be neglected by introducing a negligible approximation error.

We set $\lambda = 100$, while we add $l_2^2$ entropy regularization with parameter $\alpha = 0.01$ to the dual objective. We optimize the latter using standard gradient ascent, stopping when the $l_\infty$ norm of the gradient goes below $10^{-5}$ or 200 iterations are reached.

**LR**    We adopt the same features as for MC. Given the set of input trajectories, we learn a multi-class logistic model to predict the next state $s'$ from a state-action pair $s, a$. That is, we learn the conditional probabilities:

$$P(s'|s, a) \propto e^{\boldsymbol{w}^\intercal \boldsymbol{\phi}(s, a, s')},$$

with $\boldsymbol{w}$ as the parameter computed by the logistic model.

**RECT**    We use $l_1$ constrained uncertainty sets computed using Hoeffding's inequality with 95% confidence level, as described in Appendix A in [9].

**FQI**    We use extra-trees with an ensemble of 50 approximators, with a minimum number of 2 samples to split a node. We run the algorithm for 50 iterations and evaluate the greedy policy with respect to the resulting $Q$-function.

**MLE**    We estimate the transition probabilities using maximum likelihood:

$$P(s'|s, a) = \frac{\#[s, a, s']}{\#[s, a]},$$

where $\#[s, a]$ denotes the number of times action $a$ was taken from state $s$ in the given trajectories, and similarly for $\#[s, a, s']$.

**Additional results**    We now investigate the sensitivity of our approach to the choice of the entropy regularization parameter $\lambda$. In order to address this point, we repeat the experiments for a fixed gridworld of size $11 \times 11$ using different values of $\lambda$, with all other parameters as before. Figure 6 shows how the expected return under the true and estimated dynamics vary as a function of $\lambda$. As expected, we notice a clear performance drop under the estimated dynamics as $\lambda$ increases, i.e.,

Figure 6: Expected returns as a function of the entropy regularizer $\lambda$

as the regularization is removed. Interestingly, the performance under the true dynamics decreases by a much smaller margin. Furthermore, the robust expected return becomes higher than the true one only for very small values of $\lambda$, i.e., when keeping high entropy becomes more important than minimizing the reward. This intuitively suggests that, using intermediate values of $\lambda$, we are able to find less conservative solutions while preserving robustness guarantees under certain assumptions on the unknown MDP. Proving this statement from a theoretical perspective would be of great practical interest and is left for future work.

### C.2 Invasive Species

We provide additional details on the invasive species experiment.

**MC** We consider 14 different features. The first 10 are binary and discretize the population in bins of width 50. Thus,

$$\phi_j(s, a, s') = \mathbb{1}\left[s \in (50(j-1), 50j]\right] \quad \forall j = 1, \ldots, 10.$$

The next 2 features encode population intervals in which the two available actions transition with very high probability according to our dynamics:

$$\phi_{11}(s, a, s') = \mathbb{1}\left[a \neq \mathrm{C} \wedge (s' < s - 30 \vee s' > s + 100)\right]$$
$$\phi_{12}(s, a, s') = \mathbb{1}\left[a = \mathrm{C} \wedge (s' > s + 30 \vee s' < s - 100)\right],$$

where 'C' represents the control action. Thus, these features are enabled whenever an action fails at transitioning to an interval containing the true next state with high probability. Finally, the last 2 features encode the (absolute) population change after an action has been applied:

$$\phi_{13}(s, a, s') = |s - s'|\mathbb{1}\left[a \neq \mathrm{C}\right]$$
$$\phi_{14}(s, a, s') = |s - s'|\mathbb{1}\left[a = \mathrm{C}\right]$$

We discretize belief states in 18 different values in $[0.1, 0.9]$ and we add entropy regularization with $\lambda = 100$. For the experiment with 50 trajectories, we add $l_1$ regularization with $\beta = 0.01$ for all feature components except $\phi_{11}$ and $\phi_{12}$ (for which $\beta = 0$). As usual when given more data, we decrease $\beta$ to 0.001 for the experiment with 100 trajectories. We run MC for at most 200 iterations, stopping when the $l_\infty$-norm of the gradient goes below $10^{-4}$.

**Other algorithms** We keep the same configuration as in the gridworld experiments for all other algorithms. For LR, we use the new feature functions.

## Footnotes

[4]Whatever action is chosen, $\tilde{\pi}(a|s) = 0.5$, and, thus, we need not enumerate all possible trajectories.

[5]Recall that our objective is to find a point with zero gradient, so that the feature expectations correctly match the sample statistics.

[6] As described in Appendix B, when using a uniform reference policy we can efficiently enumerate all reachable belief states in a horizon $T$.