[Reviews · NeurIPS 2018]

Reviewer 1



The authors consider distributionally robust finite MDPs over a finite horizon. The transition probabilities conditionally to a state-action pair should remain at L1-bounded distance from a base measure, which is feasible as being generated using a given reference policy. This is a nice idea. A few comments are mentioned next. * One aspect I did not understand is how the reference policy should be chosen. Related to that question, why the requirement of staying "close" to this policy would be beneficial. In safe RL for instance, we could want to remain close to "safe" states, but I can't make the connection with this and the identification of a single "safe" reference policy. In the first experiment, (line 253) the authors use a uniform policy as the reference. In the second experiment, there are only 2 actions and the authors use a state-independent randomized policy (line 301). What is the rationale? Is the choice of a "uniform" reference policy specific to these two problems? * How should we read the results of Table 1, when p varies ? Is there a value of p that leads to a better policy? In which sense? * I liked (line 274) that the authors provided a sensitivity analysis of the stepsize they use to discretize a [0,1]-valued extra state variable. * What is the exact definition of the softmin in equations (6) and (11) * There are too few details to understand the benchmark policies such as LR, RECT, MLE to be able to reproduce the experiments (depicted in Figure 1). I would think that those methods have more parameters than those given in the supplementary material. * (line 225) "converges when the statistics match", but "when statistics match" is unclear, can the authors explain. * Section 2 introduces the notion of "Directed information theory for processes", but I interpret this as describing the fact that policies should be in the class of progressively measurable processes in discrete-time -- which is a normal math framework for history-dependent policies. * The significance is somewhat difficult to assess because the test problems seem relatively low-dimensional, which, granted, facilitates benchmarking, but does not allow to demonstrate scalability of the proposed method.

Reviewer 2



Summary: The authors propose a robust MDP formulation where the uncertainty set contains all models that satisfy a constraint based on features extracted through sample trajectories of a reference policy. The formulation allows coupling across state-action pairs such that it can address some of the shortcomings in the rectangular model. An algorithm for finding the optimal robust policy is provided and is evaluated on a synthetic and a real-world problem. This formulation seems interesting and novel to me, but I have several main concerns, mostly about the clarity of the presentation: 1. Since the major claim is that the formulation allows coupling across state-action pairs, the authors should at least illustrate and motivate how kappa and the features can be used in general, perhaps by expanding section 3.1 2. The author claims that Eq.(5) is more general than rectangular constraints. This implies that all rectangular constraints can be expressed through kappa. This doesn't seem obvious to me, perhaps the authors can show how a general s,a-rectangular constraints (say, uncertainty sets defined with max-norm balls or non-convex sets) can be translated to (5) through the features. 3. I notice that \bar{\pi} in the regularizer in (5) disappears in (6). It is not obvious why, a little remark on that would be helpful. 4. In the gridworld experiment, it is not clear to me how generalization to unseen state/actions happens through the chosen features. The authors mention 10 features in section B.1. It seems that these features only encode whether an action succeeds but not the next state itself. What is the dimension of phi here? Are the 10 features replicated for every state-action-next-state combination or a nominal model is required? What is the reward function? I assume -1 per step, but then, with N=10 and T=2N, this cannot generate a return of -200. Overall, the work seems promising but it lacks clarity in its present form.

Reviewer 3



This paper presents a procedure for computing robust policies in MDPs where the state transition dynamics are unknown, but can be estimated only through a set of training trajectories. Unlike prior work, which creates bounds the true dynamics within uncertainty sets on a state-by-state basic, this approach uses a global feature matching approach to constrain the true dynamics. The paper presents a casaul entropy optimization problem to find a robust policy and a dynamic programming procedure to compute its gradient. Great paper, pleasure to read, well written and easy to follow. There are a few important issues that have not been addressed, but I believe the paper stands well on its own as-is. There is little discussion about how one might go about selecting the state transitition features. This is obviously important for anyone who wishes to use this approach, though those familiar with inverse optimal control will have some understanding of this. Of more importance, is the sample complexity both preceived and theoretical. \tilde c should converge like 1/sqrt(N), but there should come a point (for a particular set of features) where the local uncertainty sets are more accurate. This is alluded to in the experiments, which are sufficient, but not overwhelmingly compelling. Belief states (and their discretization) seem like a detail that can somehow be polished or removed as they only appear in computing the gradient, not the final policy. Perhaps this can be accomplished by solving the saddle-point problem directly. Minor detail: Some care needs to be given to ensure gradient descent converges for maximum entropy problems. Without, e.g., L2 regularization on the parameter vector, the weights may not converge (as no solution may exist) even though the policy will usually end up being sensible.